# Living with Advanced Breast Cancer: A Descriptive Analysis of Survivorship Strategies

**DOI:** 10.3390/jcm11143992

**Published:** 2022-07-10

**Authors:** Michael Grimm, Lindsey Radcliff, Mariann Giles, Ryan Nash, Erin Holley, Shannon Panda, Lynne Brophy, Nicole Williams, Mathew Cherian, Daniel Stover, Margaret E. Gatti-Mays, Robert Wesolowski, Sagar Sardesai, Preeti Sudheendra, Raquel Reinbolt, Bhuvaneswari Ramaswamy, Ashley Pariser

**Affiliations:** 1Division of Medical Oncology, Stefanie Spielman Comprehensive Breast Cancer, The Ohio State University, Columbus, OH 43210, USA; michael.grimm@osumc.edu (M.G.); lindsey.radcliff@osumc.edu (L.R.); shannon.panda@osumc.edu (S.P.); lynne.brophy@osumc.edu (L.B.); nicole.williams@osumc.edu (N.W.); mathew.cherian@osumc.edu (M.C.); daniel.stover@osumc.edu (D.S.); margaret.gatti-mays@osumc.edu (M.E.G.-M.); robert.wesolowski@osumc.edu (R.W.); sagar.sardesai@osumc.edu (S.S.); preeti.sudheendra@osumc.edu (P.S.); bhuvaneswari.ramaswamy@osumc.edu (B.R.); 2Family Medicine, The Ohio State University, Columbus, OH 43210, USA; mariann.giles@osumc.edu; 3Center for Bioethics and Medical Humanities, The Ohio State University, Columbus, OH 43210, USA; ryan.nash@osumc.edu; 4Nutrition and Dietetics, The Ohio State University, Columbus, OH 43210, USA; erin.holley@osumc.edu; 5Internal Medicine, The Ohio State University, Columbus, OH 43210, USA; raquel.reinbolt@osumc.edu

**Keywords:** metastatic breast cancer, side effects, metavivorship, breast neoplasms

## Abstract

Survivors of advanced breast cancer (ABC), also known as metavivors, are often left with fewer treatment options in the landscape of a cure culture. Metavivors have unique psychosocial and physical needs distinct from patients with early-stage breast cancer. This analysis delves into side effects commonly experienced by patients with ABC, such as fatigue, anxiety, and cardiotoxicity; how these side effects impact caregiver support, financial toxicity, emotional strain, and spiritual and emotional distress; as well as current strategies for mitigation, including nutrition, exercise, and participation in clinical research. Overall, this analysis is a mandate for additional research to explore novel treatments and implement strategies to maintain and improve patients’ quality of life.

## 1. Introduction

Breast cancer is the most common type of cancer in women, with a lifetime incidence of 12.9%. Additionally, it affects a small percentage of men. There were an estimated 281,550 new cases diagnosed in 2021 [1]. Of these patients, 6% will be diagnosed with de novo or recurrent advanced breast cancer (ABC). Historically, this group has faced a 5-year survival of just 29% [2]. Furthermore, 30% of those diagnosed with early-stage breast cancer will develop metastatic disease [3]. The short life expectancy for patients diagnosed with ABC has led to their exclusion from survivorship services frequently offered to patients with early-stage, localized breast cancer [4]. ABC survivors are left with fewer options in the landscape of a cure culture where treatment options aimed at eradicating disease are the main focus of treatment, research and funding. However, those living with ABC have unique psychosocial and physical needs distinct from patients with early-stage breast cancer, and they merit recognition [5]. The National Institutes of Health (NIH), American Society of Clinical Oncology (ASCO), and other entities have recognized that this limited focus has resulted in a dearth of research in the area of “metavivorship”, a focus on the well-being of patients with advanced cancer [6]. Novel treatments for this population have dramatically improved survival, particularly for patients with hormone receptor-positive and HER2-positive breast cancer [7]. It is critical that we consider the various side effects patients with advanced breast cancer experience in relation to their treatment. Furthermore, research has shown that supportive care, including early involvement of palliative care, improves outcomes for patients with metastatic cancer [8,9,10]. The results of these studies counter the conventional practice of providing palliative care only in end-of-life situations. They highlight the importance of conducting clinical research on the lived experience of patients with ABC. To this end, researchers must seek to mitigate or remove any barriers to participation patients may experience.

As these clinical trials are ongoing, it is important that clinicians be aware of adverse effects commonly reported by patients with ABC. The goal of this analysis is to explore the following common side effects experienced by patients being treated for ABC: fatigue, anxiety, and cardiotoxicity. Additionally, we will review the role of nutrition and physical activity counseling to mitigate deleterious side effects of ABC treatment. Lastly, we will explore additional areas of need including caregiver support, financial toxicity, emotional strain, spiritual and emotional distress, as well as long-term immunotherapy effects.

## 2. Discussion

### 2.1. Known Side Effects

#### 2.1.1. Fatigue

Fatigue can be debilitating for patients with ABC. A meta-analysis by Al Maqbali et al. of 129 clinical trials conducted between 1993 and 2020 estimated that 60.6% of patients with advanced breast cancer experienced fatigue [11]. Fatigue is common among those who have been treated with chemotherapy, immunotherapy, or hormone therapy [12], and can be associated with the site of metastasis [13,14]. While fatigue is inherently difficult to quantify, studies employ questionnaires beginning at the time of treatment and continuing at fixed intervals afterward to identify chronic treatment-related symptoms. These studies show that 30–60% of patients report moderate to severe fatigue during treatment, and up to a quarter of them still experience symptoms 10 years after treatment [15]. Studies have also uncovered correlations between reported fatigue and overall lack of motivation as well as negative mood [16]. Studies focused on patients with ABC are more limited but suggest that effective treatment strategies may be distinct from those known to be effective for patients with limited stage disease. For example, a trial conducted by Poort et al. evaluating patients with ABC found greater benefit with cognitive behavioral therapy (CBT) compared with exercise and usual care [17]. CBT in this study addressed patients’ sleep; coping strategies; self-efficacy; physical, social, and mental activities; and social support. These findings underscore the need for specialized treatment for ABC.

#### 2.1.2. Anxiety

Anxiety among patients with metastatic cancer is an area frequently analyzed. Studies have found that those diagnosed with depression or anxiety have worse outcomes. Wang et al. performed a systematic review and meta-analysis of 2.6 million patients with any cancer diagnosis by assessing reports of anxiety/depression and measured outcomes. This analysis revealed higher cancer incidence, poorer survival, and higher cancer-specific mortality among patients with a diagnosis of anxiety and depression [18]. An Australian study of 227 patients with ABC found that over a third experienced anxiety, depression, or both, and 17% of these patients were ultimately diagnosed with an anxiety disorder [19]. Dobretsova and Deraskan found a link between anxiety, traumatic stress and decreased cognitive function. Recommending social support is an important intervention in buffering against these effects [20]. Given the broad availability of effective treatments for anxiety and depression such as psychotherapy [21], mindfulness [22], and anti-depressant medications [23], it is critical that patients are regularly screened using multimodal assessments (questionnaires, GAD10, PHQ, etc.) and subsequently referred for appropriate mental health services [24]. Unfortunately, relatively few studies have evaluated treatment for anxiety among patients with advanced cancer. Sakamoto and Koyama evaluated current medical management strategies and suggest that the medication quetiapine could be a better alternative to the more commonly prescribed benzodiazepines or selective serotonin reuptake inhibitors (SSRIs) [25]. Current guidelines therefore recommend a personalized and stepped model for treatment delivery [26].

#### 2.1.3. Cardiotoxicity

Several of the antineoplastic therapies commonly utilized to treat breast cancer are known to be cardiotoxic. For example, patients diagnosed with HER2 overexpressing breast cancer typically receive trastuzumab/pertuzumab, a treatment regimen associated with impaired heart function [27]. Studies have also reported adverse cardiac events after initiation of anthracycline chemotherapies such as doxorubicin [28]. Treatment with immune checkpoint inhibitors (ICIs) has also been associated with cardiotoxicities [29], though a consensus has not been reached [30]. Cancer therapy-related cardiac disease (CTRCD) is underestimated because of a lag in symptoms, which can present themselves up to 5 years after treatment [31]. As survivorship extends the lives of patients, managing the cardiovascular health of breast cancer survivors will come to the forefront of their care plan. Strategic surveillance methods, such as periodic left ventricular ejection fraction (LVEF) assessments [32], or novel risk assessment scoring tools like the one developed by Kozhukov et al. [33], must be employed to ensure proper identification of these side effects.

Studies evaluating alternative and novel agents are ongoing. With greater recognition of cardiac toxicities, improved screening and management is now possible. Studies suggest that beta blockers and neurohormonal blockades may still be the best available treatments against cardiac function deterioration [34]. The study of these strategies has led to the development of an entirely new field within cardiology that can serve as a resource for cancer survivors at highest risk of cardiotoxicity: cardio-oncology. Ideally, the provision of this resource allows all patients to receive standard of care without compromising their cardiac health [35]. Many of the side effects outlined above have known remedies; unfortunately, patients are not always appropriately referred to address them. 

The side effects patients experience are often a direct result of social determinants of health. Patients face barriers such as systemic racism, healthcare discrimination, implicit bias, sexism and lack of access to care. Consideration of these barriers is critically important to ensure that all patients have equitable access to care. The Living Well with Advanced Breast Cancer (LWABC) clinic created at Stefanie Spielman Comprehensive Breast Center and discussed in depth later in this review was developed to enhance supportive care and patient education.

### 2.2. Risk Reduction

A lifestyle that includes regular exercise and a diet of fruits, vegetables, whole grains, and omega-3 fatty acid-rich foods is correlated with better quality of life, favorable health outcomes, and decreased mortality [36]. The availability of such foods is demonstrably lower in areas known as food deserts, where inhabitants lack access to healthy and affordable food [37]. One study found that while the number of food desert communities has declined since 2010, 5.6% of the population still live in a low-access area, with 30% more non-white residents than white residents affected [38]. Acknowledging this racial disparity is critical in providing compassionate care to non-white cancer survivors.

While few studies have evaluated nutrition independent of physical exercise, several have assessed quality-of-life differences between healthy diets and a traditional American diet of processed and/or high-fat foods [39]. Analyses evaluating outcomes for breast cancer survivors based on diet and body mass index (BMI) have found an association between healthy eating and decreased all-cause mortality [39,40,41,42]. Along those lines, other healthy habits such as limiting alcohol intake and quitting smoking are important for improving outcomes [43]. However, prospective controlled trials are necessary to define outcomes more clearly among patients with ABC.

In studies focused solely on outcomes of patients with ABC, regular exercise was correlated with increased survival with varying levels of statistical significance [44,45]. Several studies have assessed quality of life and improved fatigue after initiation of regular exercise. Even seated exercise, performed regularly, resulted in slower total decline and less fatigue [14]. A systematic review conducted by Heywood et al. reviewed 25 studies evaluating exercise for patients diagnosed with advanced cancer. Most of these studies found improvements in physical function, fatigue, body composition, sleep quality, and psychosocial function. Survival outcomes were not found to significantly improve in any of the 25 studies evaluated [46]. The potential positive impact of physical activity on quality of life supports further research and study focused on patients with ABC. In order to promote equity, we must also consider the systemic inequalities along racial and economic lines such as the availability of exercise facilities, and time outside of work and family obligations that can be devoted to exercise [47].

### 2.3. Areas of Need for Future Study

Despite many advances in breast cancer treatment and survivorship strategies, our literature search revealed areas where additional work is needed.

#### 2.3.1. Caregiver Support

Family-centered care is key to the health of patients with metastatic breast cancer. Emotional support is important for the well-being of all people, and vital for patients with ABC. It is clear from studies focused on caregivers that they have unique needs, and the tremendous weight of their responsibilities as caregivers can lead to negative outcomes such as lower quality of life and impaired workplace productivity [48,49]. Northouse and others found that caregivers who employed maladaptive coping strategies such as alcohol use, affected the patient’s quality of life [50]. Caregiver quality of life seems to benefit from patient referral to palliative care provided early in treatment [51]. Lack of access remains a key barrier. For example, qualitative studies focused on caregivers of patients living in rural areas have confirmed caregiver demand for palliative care services and highlighted lack of access to both in person and telehealth options [52]. Maslow, Allicock and Johnson’s study of 41 Black cancer survivors and their caregivers consisted of a focus group to discuss cancer survivor and caregiver needs. Both survivors and caregivers expressed their concern about the availability of culturally appropriate support services [53]. There is a great need for community-based support structures, and it is healthcare centers’ duty to explore ways to provide these services. Caregivers are an important part of the care team and should be included when considering treatment for patients with ABC. Deeper research into how best to support caregivers is warranted.

#### 2.3.2. Financial Toxicity

There are substantial financial costs associated with any major illness, and cancer is no exception. Economically disadvantaged patients are not only lacking monetary wealth to overcome financial toxicity (FT) but are also less likely to have knowledge of financial fail-safes such as those listed in Table 1. FT can lead to decreased quality of life, and general cancer-related distress [54]. A recent study by Wan et al. evaluated the financial toxicity among patients with ABC by stratifying groups using an 11-item survey dubbed the Comprehensive Score for Financial Toxicity (COST). The groups most frequently experiencing FT included Black patients, hormone receptor-negative patients, low-income patients, and unmarried patients [55]. Lifelong continuous treatment for ABC continues to be as debilitating financially as it is medically. Expanded research and financial assistance programs directed toward patients with ABC will be crucial moving forward.

#### 2.3.3. Distress—Spiritual and Emotional

Spirituality can be an important aspect of a patient’s ability to cope with a diagnosis of ABC. Patients experiencing spiritual distress may find benefit in resources such as coordinated visits with a chaplain or other faith leader. It is in the purview of a holistic clinical approach to make these visits accessible. Several studies have reported correlations between spiritual engagement and improved health status, such as increased white blood cell count [56], greater neuroimmune activity [57], and less fatigue [58]. While these findings pose interesting comparisons to the placebo effect, what is important is that a sense of meaningfulness should be fostered given that it can bolster quality of life [59]. This highlights the role of spirituality in the well-being of patients with ABC. Patients find meaning in their lives in a variety of ways. It is important to consider a multitude of options to bolster patients’ coping skills, including mindfulness, meaningfulness, and spirituality. Spiritual distress related to cancer diagnoses remains an area warranting closer analysis, and recent studies have shown that patients care about religious and spiritual issues and are comfortable discussing them [60].

Emotional distress is a primary inhibitor of a patient’s quality and enjoyment of life. While not quite synonymous with “fear of the future” [61], distress is often a result of helplessness stemming from a metastatic cancer diagnosis [62]. Left unmanaged, emotional distress can lead to decreased quality of life and is associated with several comorbidities [63]. Studies show that emotional distress can manifest in various forms including depression and anxiety, but also existential dread and loneliness [64]. One study found that close to a third of patients with advanced cancer experienced mood disorders [65]. It has also been found that patients who engaged in active coping with a support person experienced fewer mood disorders [66]. While quality of life (QOL) surveys are ubiquitous in the study of patients with breast cancer, a more specialized and holistic analysis tailored to ABC would be valuable in helping future patients cope with their diagnosis.

#### 2.3.4. Long-Term Immune Therapy Effects

Immunotherapy is standard of care for patients with triple-negative breast cancer and will likely play a larger role in the treatment of all cancers moving forward. As treatment with immunotherapy becomes increasingly commonplace, studies evaluating immune-related adverse events (irAEs) will be critical. Patients who experience a complete response to immunotherapy treatment and achieve long-term survival will likely have unique long-term needs resulting from their treatment. These may include treatment for new chronic conditions such as type 1 diabetes, Addison’s disease, hypothyroidism, and others. IrAEs such as fatigue and anxiety present complex needs, and further study is needed to address their treatment.

#### 2.3.5. Education

Patients with ABC have unique psychosocial, physical, and treatment needs distinct from those patients with early-stage breast cancer [67]. Previous studies illustrate that clinical trials are foundational to the improvement of healthcare delivery. However, clinical trials only succeed when patients have access to the opportunity and consent to become involved. Furthermore, inequities in clinical trial focus related to supportive care screening and lack of diversity in participant population further fuel a cycle of inequality.

As such, educating all eligible patients about clinical trial opportunities is key. The education should include what clinical trials are, why they are standard of care, and when they should be considered. 

Studies of the ABC population are more often focused on novel therapeutics, rather than symptom experience or quality of life. Unfortunately, supportive care for patients with ABC has not been an area of focus despite their unique needs. Additional research must be done to better understand this special population and to identify strategies to support them during diagnosis, treatment, and palliation. Enhanced education is a documented need among patients with ABC [68], and should be viewed as a call to action for clinicians to provide educational services tailored to these patients.

### 2.4. Living Well with Advanced Breast Cancer (LWABC) Clinic

Due to past and ongoing treatments, many patients with ABC have persistent physical and psychological symptoms, which if addressed are amenable to treatment and optimization. Furthermore, prognostic uncertainty and limited understanding of their disease can limit patients’ ability to make informed treatment decisions. 

The Living Well with Advanced Breast Cancer (LWABC) clinic, established in 2017, is a comprehensive, personalized consultative visit currently offered to patients with ABC at Stefanie Spielman Comprehensive Breast Center, a specialized care facility associated within The Ohio State University Comprehensive Cancer Center (OSUCCC; Columbus, OH). Patients and caregivers meet with specialists in breast medical oncology, integrative medicine, palliative care, nutrition, and nursing. They receive disease-specific education, including a discussion on biomarkers, tumor biology, the role of research, and supportive care services.

The structure of the LWABC clinic allows providers to dedicate time to elucidate patients’ needs in ways impossible in a traditional healthcare setting. The LWABC clinic visit is broken into five consultations conducted over a 2.5-h block, and each modality approaches an important aspect of ABC care (Table 2).

#### 2.4.1. Patient Experience with LWABC

The LWABC clinic has been popular among patients. Over 95 patients have been served to date. Surveys were distributed to patients seen in the LWABC asking them to assess the quality of the clinic (scale: 1 = very poor, 2 = poor, 3 = fair, 4 = good, 5 = very good) and likelihood they would recommend the clinic to others (scale: 1 = will not recommend, 2 = unlikely, 3 = unsure, 4 = likely, 5 = very likely), and results are listed in Table 3. Of the 46 patients analyzed, the mean overall experience was 4.83, with 72% reporting a 5/5 experience with each consultation. Often, patients were subsequently referred for additional supportive care services, most commonly for physical therapy, psychosocial oncology, social work, occupational therapy, and palliative medicine.

Below are illustrative quotes of feedback received via surveys to date:“Even though I’ve been in Cancerland for 5 years (and thought I knew a lot), I learned quite a bit.”“I found it to be very helpful and it also put me in a better place emotionally.”“I am now armed with accurate and useful information and am less uncertain.”“Excellent program. I loved the time spent with me by each professional. Very good information & well presented!”“It makes you feel like a person and not a number.”“I feel very blessed to be able to have my family with me. It was very helpful for us to learn together, and it gave us a starting point for some of the more difficult discussions. Also, discussing the cancer as a journey was so helpful to me and my family because it helped us to understand we will be able to plan for things.”

#### 2.4.2. Current Limitations 

The LWABC clinic was designed to meet educational and supportive care needs of patients with ABC [6,69]. However, patients with ABC constitute a diverse population. Personalization of the visit to include or exclude portions of the visit is not currently possible due to time and scheduling constraints. Thus, additional visits are required to meet with specialists in other supportive care domains such as psychosocial oncology or cancer rehabilitation. The visit is currently 2.5 h in length. It was structured in this way to allow for a single virtual or in-person visit with multiple specialists. However, this time commitment is likely prohibitive for patients with competing commitments such as work or caregiving responsibilities. To enhance the patient experience, a nurse navigator was recently hired. Future goals include implementing a pre-visit needs assessment, enhancing patient feedback, and augmenting community referral options.

## 3. Conclusions

The last five years have seen incredible advances in the treatment of ABC, and it is heartening that long-term survivorship services for patients are evolving to meet their needs. Further research is needed to evaluate novel treatments and implementation of strategies to maintain and improve patients’ quality of life. Indeed, patients left behind by traditional survivorship services deserve individualized attention befitting their unique health and wellness concerns. Our model has been successful in meeting these needs, and in offering a supportive space for patients to explore their survivorship and discover ways to cope with the difficulties of life with ABC. This review provided a summary of common adverse effects and known unmet needs and outlined our innovative survivorship clinic dedicated to advancing education and supportive care services for patients with ABC.

## Figures and Tables

**Table 1 jcm-11-03992-t001:** Several financial assistance programs can allay financial strain resulting from extensive cancer therapy. However, many are geared toward short-term treatments provided for early-stage cancers. Here, we list resources tailored to assist with management of long-term cancer treatment.

Organization	Website	Phone	Assistance
American Cancer Society	www.cancer.org (accessed on 22 April 2022)	800-227-2345	Reimbursements for costs associated with cancer treatment
Breast Cancer Assistance Fund	https://breastcanceraf.org (accessed on 22 April 2022)	866-413-5789	Need-based financial assistance for non-medical costs of getting a patient to treatment and other living expenses that may be incurred
Cancer Care	www.CancerCare.org (accessed on 22 April 2022)	800-813-HOPE	Counseling, education, support groups, need-based financial aid for treatments
Cancer Care Co-Payment Assistance Foundation	www.cancercopay.org (accessed on 22 April 2022)	866-552-6729	Copay assistance
Cancer Financial Assistance Coalition	https://www.cancerfac.org (accessed on 22 April 2022)	not applicable	Coalition of financial assistance organizations joining forces to help cancer patients experience better health and well-being by limiting financial challenges.
Cancer Supportive Care	http://www.cancersupportivecare.com/drug_assistance.html (accessed on 22 April 2022)	Not applicable	Listing of pharmaceutical drug assistance programs with contact information.
The Health Well Foundation	www.healthwellfoundation.org (accessed on 22 April 2022)	800-675-8416	Need-based financial assistance with coinsurance, copays, premiums, and deductibles.
Needy Meds	www.needymeds.com (accessed on 22 April 2022)	Not applicable	Non-profit organization with a database of patient assistance programs to include drug discount programs and state assistance programs.
Susan G. Komen	www.komen.org (accessed on 22 April 2022)	877- 465-6636	Need-based financial assistance with treatment payments
Patient Access Network Foundation	www.panfoundation.org (accessed on 22 April 2022)	866-316-7263	Need-based financial assistance with copays, deductibles, and medications.
Partnership for prescription assistant	https://www.phrma.org/patient-support (accessed on 22 April 2022)		Non-profit organization sponsored by Pharmaceutical Research and Manufacturers of America’s (PhRMA), civic groups, and patient advocacy organizations. This group is dedicated to helping patients find free or low-cost brand-name prescriptions.
Patient Advocate Foundation on CoPay Relief	www.copays.org (accessed on 22 April 2022)	866-512-3861	Need-based financial assistance with copays for prescription drugs
Remember Betty	http://rememberbetty.com (accessed on 22 April 2022)		Helps minimize the financial burden associated with breast cancer for patients and survivors so they can focus on recovery and quality of life.
RxHope	https://www.rxhope.com/ (accessed on 22 April 2022)	Not applicable	Rx Hope advocates for patients and helps them navigate available assistance programs.

**Table 2 jcm-11-03992-t002:** The LWABC clinic is broken into five sections, each addressing a different aspect of ABC and focused on overall well-being.

Provider	Education	Time
Breast Medical Oncologist	Overview of advanced breast cancer and importance of supportive care Explain diagnosis and treatment approach, including discussion of cutting-edge technology and therapy. ○ CDK4/6 inhibitors, antibody drug conjugates, etc. Explain role of cancer care team and role of patient/caregiver. Assess individual concerns.	45 min
Palliative Medicine	Overview of pain, symptom management, and palliative care services Address patient health and well-being using a bio-psychosocial-spiritual model. Discuss common symptoms and treatments. Address fears and approaches to living with cancer. ○ Emphasize that palliative care is not merely end-of-life care. Review special considerations for advance directives with advanced breast cancer.	45 min
Integrative Medicine	Review of diet, lifestyle, supplements, and other integrative modality (for example: acupuncture, massage) options that could complement breast cancer therapies or help the patient relieve symptoms.	30 min
Registered Dietician	Explain benefits of following a plant-based diet, including increased immune response and weight control. Encourage eating with a balanced plate and include a wide variety of fruits, vegetables, whole grains, beans, nuts, seeds, and lean proteins. Encourage the use of whole foods over processed foods to increase nutritional value. Educate patients about common nutrition myths and misinformation.	30 min
Nurse Navigator	Review referrals/recommendations placed during visit. ○ Virtual patient support groups ○ Advice for talking to children about cancer ○ Symptom management teaching Review additional resources available, including financial assistance, lifestyle resources, educational options, etc.	10 min

**Table 3 jcm-11-03992-t003:** LWABC survey analysis.

Consultations to Date	Number of Surveyed Participants	Mean Overall Experience Reported (0–5)	Participants Who Reported 5/5 Experience
95	46	4.83	33

## Data Availability

Not applicable.

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
