# Peer review of "Living with Advanced Breast Cancer: A Descriptive Analysis of Survivorship Strategies"

_jcm, 2022, doi:10.3390/jcm11143992_

Round 1
Reviewer 1 Report
The topic of survivorship for patients with advanced breast cancer is extremely important. However, after reviewing this manuscript closely, I have significant concerns about bias in both the methods of this paper and the design of the described program.
Overall comments:
1. Please clarify the strategy that was used to determine what topics were covered and what articles were cited. For example, was any systematic process used to curate strategies and evidence?
2. Please discuss internally the process by which you constructed a review on survivorship for people with advanced breast cancer that does not offer any analysis of social or structural context, and yet makes sweeping generalizations about Black patients and their social support. This causes me great concern.
Introduction:
1. Please remove judgemental language such as "bleak" before "prognosis." Simply describing this as "short life expectancy" or something similar is clearer.
2. More should be explained about "cure" culture and what is meant by this.
3. "Statistics" in unnecessary in "survival statistics." You may say simply "survival."
4. "As patients with advanced disease live longer" - the listed areas of focus were always necessary. Please consider revising.
5. "Furthermore, research has shown that supportive care, including early involvement of palliative care improves outcomes for patients with metastatic cancer further highlighting the need to advance supportive care for patients with ABC, and encourage their participation in clinical research [7]." - This sentence is not clear. Temel's paper is from over a decade ago, so the urgency of doing this now is a bit lost. Additionally, why is clinical research participation needed? Considering diving into two sentences and clarifying the goal of each.
6. "Given strides within the field of survivorship and supportive care" - this is an abrupt transition, what are the strides? Please cite and consider a clearer transition from the above paragraph.
7. "In addition, research of caregiver support or lack thereof, financial toxicity, emotional strain, spiritual and emotional distress, and long-term immunotherapy effects is warranted. Optimization of nutrition and physical activity remain underutilized and effective strategies to mitigate deleterious side effects of ABC." - This is another sentence that is overly long and not clear in its intent. It also does not have the appropriate citations.
8. There is not sentence introducing the goal of this paper.
Discussion:
1. "Fatigue can be debilitating for patients with ABC, and is often related to the site of metastasis, which can present in the bones or viscera, and varies by treatment regimen" - fatigue is certainly debilitating, but is multifactorial, not particularly related to metastasis site specifically, though does vary certainly by treatment. Please clarify and offer additional citations on cancer-related fatigue.
2. "antineoplastics and immunotherapy" - immunotherapy is antineoplastic. Please revise.
3. Anxiety - what strategies have evidence? Many are listed.
4. Duloextine is a treatment of CIPN related pain, please clarify.
5. Please remove "armamentaria" - this does not increase clarity.
6. Line 130-134 is out of place.
Figure 1:
- This figure does not correlate with the numbers cited in the primary literature from the text. It may be most helpful to remove the figure.
Risk reduction:
1. "The two primary tenets of health maintenance are good nutrition and regular exercise." - There are not two primary tenant of "health maintenance" and "health maintenance" isn't a particularly precise term.
2. "healthy diets" is extraordinarily subjective and not particularly meaningful.
3. "Analyses evaluating outcomes for breast cancer survivors based on diet and BMI have found an association between healthy eating and decreased all-cause mortality." - This lacks consideration of the tremendous social context that structures access to food, recreation, and that is colinear with poor versus favorable outcomes.
4. Please revise this entire section to reflect social risk factors and structural determinants of health, particularly racism.
Areas of Need for Future Study:
1. "Despite many advances, we are aware of several gaps in the literature" - how did you determine these gaps?
2. "carriers the burden of their loved one’s progressive disease" - please find other language to describe what you're getting at here.
3. "Black patients with breast cancer encounter additional complex barriers to emotional health services." - what does this mean?
4. "Discussion of cancer is culturally taboo in the black community, and supportive care services are seldom utilized [45]." - You have offered a single citation for a sweeping generalization for an entire, diverse group of people. Please remove this sentence. It's very concerning that this statement appears in your work and sincere self-reflection on what this was considered appropriate is required.
Survey
1. What were the demographics of the patients that were surveyed? With the above statements, it's really concerning that there is bias in program and who is designed for.
Reviewer 2 Report
Some minor comments to the paper:
1) In the introduction section, long-term survivorship is not limited to HR+ patients, but also to HER2+: please comment, considering the new data released at ESMO 2021 and ASCO 2022
2) Fatigue: most of the data are reported and discussed with regards to EBC patients: please, detailed and present more data on ABC ones
3) Don't agree to put CIPN into this topic: CIPN is mostly present in early stage patients, treated with taxanes, rather than in ABC ones. Try to better discuss, or remove the section
4) Cardiotoxicity is discussed for and limited to anti-HER2 agents: considering that cardiotoxicity is often reversible, I suggest to discuss here the long-term effects of anthracyclines and provide suggestions for the best surveillance
5) No section is present on working limitations for ABC: I suggest to add a section on this topic
